# Potential Therapeutic Use of Stem Cells for Prion Diseases

**DOI:** 10.3390/cells12192413

**Published:** 2023-10-07

**Authors:** Mohammed Zayed, Sung-Ho Kook, Byung-Hoon Jeong

**Affiliations:** 1Korea Zoonosis Research Institute, Jeonbuk National University, Iksan 54531, Republic of Korea; mzayed2@vet.svu.edu.eg; 2Department of Bioactive Material Sciences, Institute for Molecular Biology and Genetics, Jeonbuk National University, Jeonju 54896, Republic of Korea; 3Department of Surgery, College of Veterinary Medicine, South Valley University, Qena 83523, Egypt; 4Department of Bioactive Material Sciences, Research Center of Bioactive Materials, Jeonbuk National University, Jeonju 54896, Republic of Korea

**Keywords:** stem cells, Creutzfeldt–Jakob disease, prion, neurodegenerative diseases, cell therapy

## Abstract

Prion diseases are neurodegenerative disorders that are progressive, incurable, and deadly. The prion consists of PrP^Sc^, the misfolded pathogenic isoform of the cellular prion protein (PrP^C^). PrP^C^ is involved in a variety of physiological functions, including cellular proliferation, adhesion, differentiation, and neural development. Prion protein is expressed on the membrane surface of a variety of stem cells (SCs), where it plays an important role in the pluripotency and self-renewal matrix, as well as in SC differentiation. SCs have been found to multiply the pathogenic form of the prion protein, implying their potential as an in vitro model for prion diseases. Furthermore, due to their capability to self-renew, differentiate, immunomodulate, and regenerate tissue, SCs are prospective cell treatments in many neurodegenerative conditions, including prion diseases. Regenerative medicine has become a new revolution in disease treatment in recent years, particularly with the introduction of SC therapy. Here, we review the data demonstrating prion diseases’ biology and molecular mechanism. SC biology, therapeutic potential, and its role in understanding prion disease mechanisms are highlighted. Moreover, we summarize preclinical studies that use SCs in prion diseases.

## 1. Introduction

The normal physiological cell-surface prion protein (PrP^C^) is highly expressed in a variety of tissues in mammalian species [1,2]. During the early embryogenesis stage, PrP^C^ plays an important role in neural development and adult neurogenesis as well [3]. Lopes and his colleagues showed that PrP^C^ stimulates polarization in synapse development in embryonic hippocampal neuron cultures [4]. PrP^C^ has been involved in a variety of functions including signal transduction [5], cell adhesion, and antiapoptosis activity [6], neuronal differentiation, neurite outgrowth [7], and proliferation and neuronal differentiation of stem cells (SCs) [8,9]. On the other hand, PrP^C^ also undergoes a conformational conversion into a misfolded β-sheet-rich structure known as PrP scrapie (PrP^Sc^). It has been reviewed that once aggregates of PrP^Sc^ deposit within brain tissue, they induce the neuropathological characteristics of prion diseases, or transmissible spongiform encephalopathies (TSEs), including vacuolization (spongiosis), neuronal death, and PrP^Sc^ deposits [10].

Scrapie, which affects sheep and goats, was the first prion disease discovered in animals. Later, cattle were diagnosed with prion disease (bovine spongiform encephalopathy; BSE); deer, elk, and moose (chronic wasting disease; CWD), minks (transmissible mink encephalopathy), and felines (feline spongiform encephalopathy) can manifest forms of the disease [11]. In humans, most cases occur sporadically due to the spontaneous misfolding of PrP^C^ into PrP^Sc^. An example of sporadic human prion disease is sporadic Creutzfeldt–Jakob disease (CJD). Jeong and Kim reviewed the examples of the genetic forms of human prion disease including fatal familial insomnia, Gerstmann–Sträussler–Scheinker disease (GSS), and familial CJD [12]. Due to the potential transmission between other species and humans, the scientific community has placed a particular emphasis on prion diseases.

No potential therapeutic agents or treatments are available for prion diseases, except those available for the inhibition of PrP^Sc^ accumulation and to decrease the conversion of PrP^C^ into PrP^Sc^ [13,14]. Unfortunately, when delivered at the late stage of the disease, these drugs have not shown in vivo efficacy [15]. Therefore, it is important to develop treatments that target prion propagation and pathogenesis. Moreover, although prion research has greatly progressed over the last few decades, many unanswered concerns remain regarding prion replication processes, cell toxicity, genetic susceptibility variations associated with prion strains, and the nature of prion strains. SCs are self-renewing multilineage differentiated cells that can be isolated from various sources of tissues including bone marrow (BM), adipose tissue, brain, blood, dental pulp, synovial fluids, and other tissues [16,17]. In addition to their significant regenerative and immunomodulatory activities, SCs are also being utilized as essential cell models to understand the mechanism of several diseases. Neural SCs (NSCs) comprise a promising key subset of SCs currently being employed as a potential therapy for prion diseases. Thus, SCs might be a good cell model to understand the mechanism of prion disease and as a promising therapy as well. In this paper, we first discuss the molecular biology of prion disease. In addition, we highlight the proposed roles of SCs as a potential in vitro model for prion diseases and their promising therapeutic application for prion diseases.

## 2. Molecular Biology and Pathogenesis of Prion Diseases

It has been concluded that prion diseases are deadly, transmissible, and irreversible neurodegenerative disorders triggered by aberrant aggregated prion proteins in a wide variety of hosts (Figure 1) [18,19].

TSE is subdivided into CJD and kuru for humans, scrapie for sheep, and BSE for cattle [18]. Prion disorders can be sporadic, inherited, or acquired and are transmissible within and between mammalian species [20]. There have been numerous cases of variants of CJD resulting from the transmission of BSE to humans [21]. The suffering of millions of animals from BSE causes major food crises worldwide. The cellular-host-encoded prion protein is an alpha-helical neuronal glycoprotein [22]. However, the role of PrP^C^ has not been entirely elucidated. Nevertheless, Westergard et al., 2007 reviewed that PrP^C^ has important physiological functions regarding cellular proliferation, adhesion, differentiation, neural development, and immune response due to its location on the cell membrane [23]. Additionally, PrP^C^ has a multimolecular signaling pathway in the neuronal differentiation process [9,24]. The globular C-terminus, which contains three alpha-helices and two short ß-strands, in addition to the unstructured N-terminus, is one of the two principal domains in PrP^C^. PrP^C^ is found in different cells such as neurons, glial cells, lymphocytes, and follicular dendritic cells [25,26,27]. PrP^C^ is mostly prevalent in brain tissue, although it is also detected in the heart, skeletal muscle, and kidney, while it is barely detected in the liver [2].

PrP^C^ undergoes a conformational change in prion diseases, resulting in a misfolded, beta-sheet-rich, and aggregation-prone variant (known as PrP^Sc^) (Figure 1), which is a partially protease-resistant isoform [18,19]. A characteristic feature of prion diseases is the formation of the aberrant isoform PrP^Sc^ of the host-encoded PrP^C^ in the central nervous system. It has been concluded that PrP^Sc^ aggregates deposit and propagate inside the brain, eventually resulting in prion disease pathologies such as neuronal vacuolation, significant apoptosis, neuroinflammation, and neurotoxicity, all of which cause neurodegenerative disorders [28,29,30]. The normal isoform PrP^C^ is protease-sensitive (PrP^Sen^), but the pathological isoform PrP^Sc^ is somewhat protease-resistant (also known as PrP^res^). PrP^Sc^ is an infectious agent without nucleic acid and contains only an abnormal conformer of PrP^C^, known as the ‘protein only hypothesis’, which affects the prion propagation and infectivity [19]. Therefore, detection of the protease-resistant core of PrP^Sc^ on immunoblotting assay serves as an accurate molecular marker and aids in the diagnosis of the presence of the infectious agent [31,32]. The immunoblotting procedure is used and validated for the detection of protease-resistant PrP^Sc^ [33,34].

PrP^Sc^ serves as a conformational template, attracting PrP^C^ for subsequent conversion, and this process is self-replicating. Prions primarily, if not entirely, have a damaging effect on the central nervous system [30,35]. In prion disorders, the conversion response is critical for neurotoxicity [35]. However, the underlying cause of neurotoxicity remains unknown. PrP^C^ is necessary for prion development, as it has been reported that PrP^C^-deficient animals are resistant to prions [36]. Moreover, it has been demonstrated that transplanting neural tissue overexpressing PrP^C^ into the brain of PrP-lacking mice produces excessive levels of PrP^Sc^ and induces prion-like disease characteristics [37]. Furthermore, neurotoxicity was reversed when the endogenous neuronal PrP^C^ was depleted in established prion-infected mice [38]. Taken together, these studies suggest that misfolded PrP^C^ elicits neurotoxicity and neurodegeneration. Therefore, strategies aimed at restoring normal function and promoting neuroregeneration should be pursued for therapeutic advancements. Unfortunately, no therapies with confirmed advantages against prion diseases are currently available.

## 3. Current Therapeutic Strategies for Prion Diseases

Treating prion disorders is an incredibly difficult task. There are currently no treatments available for prion disorders [39]. Since the disease proceeds rapidly and is always deadly, there is an urgent need for medicines that target prion pathogenesis. The finding that misfolded proteins are implicated in other neurodegenerative disorders is favorably influencing the field of prion drug research [40]. Effective treatment for prion diseases should either prevent the formation of misfolded proteins or mitigate the development of neurotoxic effects. In this section, we briefly summarize the advancements that have led to improvements in the drug development process for prion disorders.

### 3.1. Target PrP^C^

One approach to combating prion diseases is to target PrP^C^. Numerous PrP^C^-targeting therapies have been developed. The utilization of the adult-onset model of PrP^C^ depletion has provided support for the concept of the efficacy of approaches targeting PrP^C^ [38]. This transgene-modulated elimination of neuronal PrP^C^ during developed prion infection allowed the animals to remain without symptoms for an extended period and resulted in the reversal of early spongiform alterations. Anti-PrP monoclonal antibodies, for example, have been shown to block the integration of PrP^C^ into pathogenic spreading prions in prion-infected cultured cells, resulting in PrP^Sc^ removal [41]. However, this technique may not be enough for treating patients whose prion disease has already developed to the extent of showing neuropathological effects.

### 3.2. Inhibit the Conversion of PrP^C^ to PrP^Sc^

Because the conversion of PrP^C^ to PrP^Sc^ is essential for prion pathogenesis, drugs that target this pathway are attractive treatment prospects for prion diseases [42]. Many elements of prion conversion might be studied, for example, compounds such as polysulfated polyanionic, polyamine, tetrapyrroles, polyene antibiotics, tetracyclic, tricyclic, and peptides [43]. Nevertheless, these drugs have minimal therapeutic impact on disease progression in vivo, poor bioavailability, and need further clinical trials [43]. Moreover, they have only been demonstrated to slow disease development when given prophylactically around the time of prion inoculation and when the infection is limited to the lymphoreticular system [44]. An effective and sensitive prion infectivity bioassay is needed for clinical diagnostics and to validate the anti-prion substances. Real-time-quaking-induced conversion (RT-QuIC) has recently emerged as a very sensitive technique for detecting PrP^Sc^ [45]. RT-QuIC is an in vitro amplification assay that enables the real-time monitoring of the aggregation activity of misfolded prion proteins [45,46]. It serves as a prescreening assay for substances that potentially prevent the aggregation development of the PrP^C^ to PrP^Sc^ such as doxycycline, carnosic acid, acridine, dextran sulfate sodium, tannic acid, curcumin, and poly(propylene imine) glycodendrimers [47,48,49,50]. Moreover, the RT-QuIC assay currently provides a validated diagnostic tool for human patients [51]. Different review opinions stated the capability of the small molecule theragnostic to combine imaging and treatment at the same time, presenting great promise to treat and diagnose in vivo prion diseases through its ability to bind with PrP^C^ and consequently prevent prion conversion [52,53,54].

### 3.3. Clearance of PrP^Sc^

It is commonly known that PrP^Sc^ aggregation in the brain induces neuronal cell death. As a result, treatments that either increase PrP^Sc^ clearance or inhibit its toxicity might be potentially beneficial [55]. While there is no change in PrP^Sc^ levels, therapeutics that suppress the unfolded protein response result in clinical improvements. Quercetin, a flavonoid molecule of the polyphenol group, has been discovered to break down prion fibrils in vitro in the battle against prion fibrils. Quercetin-like molecules bind to prion fibrils and reduce the β-strand content by transforming certain β-strands into loop and helical structures, causing the fibril structure to disaggregate [56]. Our study recently showed that clonidine-treated prion-infected mice displayed a significant clearance of accumulated PrP^Sc^ by stimulating the glymphatic system, the brain’s perivascular waste-clearing mechanism [33]. However, clonidine did not completely cure the prion-infected mice.

## 4. Mesenchymal SCs (MSCs)

Hoang et al., 2022 reviewed that MSCs have unique abilities such as self-renewal, differentiation capability, immunomodulation, and migration to injured tissue, making them excellent candidates for the treatment of musculoskeletal, neurological, eye, oral, and systemic disorders [57]. Embryonic SCs (ESCs), induced pluripotent SCs (iPSCs), and adult MSCs are the three major types of SCs used for treatment purposes (Figure 2).

The pluripotent capacity and ethical concerns (use of germ cells) are a major challenge for ESCs (reviewed by Lo and Parham) [58]. iPSCs do not exist in nature, but they are reprogrammed in culture by the necessary expression of factors essential for managing the crucial properties of ESCs [59]. However, genetic mutations, tumorigenesis, immunogenicity, and epigenetic abnormalities are the major concerns of the iPSC type [60]. The third type of SCs is adult MSCs, which are isolated from numerous types of adult organs and tissues such as BM, adipose tissue, dental pulp, brain, cartilage, synovial fluid, blood, and other sources [61,62]. Friedenstein and his colleagues pioneered the use of MSCs [63]. Since then, significant advances have been made in describing them. MSCs are now a hot topic in cell therapy and bioengineering research. They are found in high numbers in BM, adipose tissues, umbilical cord, peripheral blood, synovial fluid, dental tissues, and placental tissues. MSCs can be grown in various undifferentiated phases to develop into highly specialized cells that produce secretory substances, enhancing tissue regeneration in the body. In recent years, extensive research has been undertaken on the extraction and characterization of MSCs derived from diverse sources. However, it has been concluded that their characterization is still a topic of discussion [64].

Considerable progress has been made in characterizing adult MSCs. According to the International Society for Cellular Therapy, the minimum criteria for identifying MSCs are (1) adhesion to the culture dish, (2) ability to form colony units, (3) trilineage differentiation into adipogenesis, chondrogenesis, and osteogenesis, and (4) positive expression of cell surface markers CD105, CD90, CD73, major histocompatibility complex (MHC) class I and negative expression of CD34, CD45, and MHC II [65]. MSCs do not express MHC II so they are less susceptible to attack by immune cells [66]. Considerable efforts have been made to regulate the microenvironment of SCs in vitro through factors such as seeding density, passage number, coating surfaces, and three-dimensional scaffolds [67]. Furthermore, various approaches, including preconditioning with biological agents and cytokines, genetic alteration, and hypoxia application have been proposed to enhance MSC characteristics [17,68].

Despite the relevance of MSCs in regenerative medicine, their usage is not without complications. Earlier research has reviewed that MSCs are very heterogeneous, comprising cells with various multipotent characteristics [69]. There is still a shortage of understanding of the molecular process involved in the identification and isolation of MSCs. The immune system perceives MSCs as invading cells, leading to immunological rejection in specific types and procedures of MSCs. Additionally, the impact of the inflamed environment on their differentiation potential still needs to be extensively investigated [70]. Nevertheless, SC transplantation is already being used to treat a variety of diseases and disorders.

## 5. Regenerative Potential of MSCs

Hoogduijn and Dor 2013 concluded that BM and adipose tissue MSCs are the generally well-known adult sources; they are also the most generally used for clinical applications due to the less invasive collection, relative abundance inside the body, and excellent tissue regeneration capabilities [71]. The properties of MSCs vary depending on the tissue from which they are collected. Adipose tissue-derived MSCs (AdMSCs) demonstrate a high proliferative capacity and resistance to the effects of aging [72], while BM-derived- MSCs (BM-MSCs) exhibit high osteogenesis and low proliferation with age [73]. Another important source is NSCs, which are self-renewing multipotent cells in the nervous system that can differentiate into neurons, astrocytes, and oligodendrocytes. Maldonado-Soto et al., 2014 reviewed that cultured NSCs have contributed to understanding the mechanisms underlying the formation of neurons and glia [74]. The distinctions among diverse populations and sources of human MSCs have been highlighted, including their biological features, surface marker expression, proliferation and differentiation capacity, immune-modulatory properties, and variances in the extracellular microenvironment [75]. Previously, it was thought that MSCs had therapeutic potential due to their capacity for differentiation. However, new research has concluded that MSCs provide biological and regenerative benefits through the release of paracrine factors [76,77,78] (Figure 3).

These paracrine factors play an important role in cell-to-cell communication, regulating cell proliferation and adhesion, and exhibiting immunomodulatory activities [79]. Due to the interaction between MSCs and immune cells, MSCs play a role for clinical purposes. MSCs exhibit immunomodulatory mechanisms through the secretion of factors such as IDO, prostaglandin E2, transforming growth factor-β, and human leukocyte antigen G5, which interact with immune cells including B and T cells, dendritic cells, and macrophages (reviewed by Gao et al., 2016) [80] (Figure 3). MSCs possess immunoregulatory properties derived from their interactions with immune cells in both the innate and adaptive immune systems. They can decrease CD8+ T cell proliferation and cytokine production [81], and increase the percentage of functionally induced CD4+CD25+Foxp3+ regulatory T cells and IL-10 secretion [82]. Therefore, MSCs are a potential treatment for progressive multiple sclerosis by inhibiting the proliferation and cytotoxicity of Natural Killer cells and increasing the production of regulatory T cells [83]. In addition to their immunomodulatory processes, MSCs release biologically active molecules such as growth factors, cytokines, chemokines, and exosomes (Figure 3). These paracrine factors are vital in suppressing cell apoptosis and fibrosis, promoting tissue healing, and stimulating wound remodeling [84]. Zayed and Iohara reviewed the different types of mature MSC release molecules such as TGF, hepatocyte growth factor, IDO, vascular endothelial growth factor, insulin-like growth factor, fibroblast growth factor, macrophage colony-stimulating factor, and cytokines (IL-6, -8, -10) [85].

Recent research has shown that the paracrine factors indicated above are released in MSC-derived extracellular vesicles (EVs). EVs serve as essential paracrine regulators of MSCs that exist in cell supernatants and play an important role in cell signaling. EVs are classified into four types based on their diameter: exosomes, microvesicles, apoptotic bodies (formed following cell death), and endosomes [86]. These EVs can encapsulate and transport various bioactive molecules including proteins, lipids, nucleic acids, and organelles to contact cells [87]. Exosomes and microvesicles appear to have essential roles as EV mediators in a wide range of physiological processes. The EV-mediated cellular communication between MSCs and a variety of target cells, including macrophages, microglia, chondrocytes, articular chondrocytes, endothelial cells, fibroblasts, pericytes, NSCs, neurons, hepatic stellate cells, and podocytes, demonstrates MSC-EVs’ therapeutic potential in immune modulation and tissue repair [88]. Moreover, while EVs and MSCs have similar therapeutic benefits, EVs have a safety profile superior to that of MSCs due to their lack of cellular content, decreased immunogenicity, and capacity to pass the blood–brain barrier as reviewed by Gowen et al., 2020 [89]. Several clinical trials have been conducted to evaluate MSC-derived EVs for the treatment of lung and kidney fibrosis, spinal cord injuries, skin injuries, osteoarthritis, and type 1 diabetes [86]. Considering all the benefits and activities mentioned above, MSCs are concluded to represent a promising cell-based treatment option for many tissue disorders [16,90,91,92].

## 6. MSCs as a Cell Model for Prion Diseases

Prion bioassays in murine mice are commonly utilized to study prion diseases [93]. These costly and time-consuming animal trials, however, are impractical for evaluating various substances possibly effective for anti-prion therapy, particularly in humans. It has become obvious, in particular, that anti-prion substances discovered using mouse prions do not display efficacy against human prion strains, since the desired drugs must be capable of crossing the blood–brain barrier [94]. The inadequate recapitulation of other features of human prion disease and the limited lifespan that prevents the development of a phenotype are among the challenges of using animal models [93]. Thus, more models are anticipated to be developed to handle the increasingly complicated concerns of prion biology. As a result, accurate in vitro paradigms for investigating prion propagation are needed to test potential therapies in a high-throughput manner [95]. Cell culture has long been employed as an effective tool for molecular biology research, and it continues to provide vital insights into processes. However, the current cell culture models for human prion disease are not of sufficient quality, leading to limitations in exploring anti-prion drugs for clinical application [13]. Prion strain-specific therapeutic effects, the emergence of drug-resistant prion strains after long-term treatment, and the difficulties of reproducing human prions in cultured cells are all obstacles [96,97,98,99,100]. A more likely explanation of the difficulty of propagating human prions in cultured cells is that human prions have a peculiar feature that makes them difficult to replicate in cultured cells [101,102]. Additionally, chromosomal abnormalities and genomic instability can give rise to cancers driven by aberrant gene expression. Hence, it is important to employ cell lines with consistent cellular phenotypes and chromosomal stability. To better understand the biological characteristics and therapeutic potential of different therapies, neural cell lines were employed as a cellular model for prion disease [103,104]. N2a neuroblastoma cells are the most cultured cell line used for the propagation of prion diseases [105]. GT1 is hypothalamic, and CAD5 catecholaminergic cells have become popular models in prion diseases as well [96]. Prion strains can potentially infect non-neuronal cells such as fibroblast cell lines (NIH-3T3, L929, and skeletal myoblasts cells) [106,107]. The microglial cell line MG20 was employed to investigate the mechanism of the host immune reaction in prion infection [108]. In addition, genetically engineered cells such as N2a#58, RK13, and NpL2 have been utilized to explore prion diseases [99,109,110]. Nevertheless, significant drawbacks of employing mouse cell lines for prion infection have been mentioned, including the cytotoxic effects associated with prion propagation and cell death after active prion propagation in primary neurons [111].

MSCs offer significant benefits over other cell lines as a cellular model for prion diseases. The following are the key benefits of using MSCs as a cell model in studying cellular and molecular biology.

Because SCs are derived from healthy tissues, they constantly exhibit normal physiological conditions.The genomes of SCs are devoid of aberrations and can be exceptionally durable [112].SCs can differentiate into a variety of cell types.SCs can produce organoids, which allow cellular processes to be investigated in the context of differentiated tissue.

It has been demonstrated that BM-MSCs express PrP^C^, which decreases with the subsequent passage [113]. PrP^res^ accumulation in MSCs of experimentally infected mice might lead to prion manifestations in the brain [114]. The presence of PrP^res^ in BM-MSCs collected from CJD patients was discovered in the same study, and it was suggested as an alternate means of diagnosis. Lyahyai and his colleagues demonstrated that MSCs from ovine peripheral blood have been isolated and characterized, expressing SC markers and differentiating into adipogenesis, osteogenesis, chondrogenesis, and neurogenesis [115]. According to the findings, MSCs can express the cellular prion protein gene (PRNP), which is increased during neurogenesis. Taken together, the influence of prion infection in monolayer-cultured ovine BM-MSCs and BM-MSC-derived spheroids was evaluated, demonstrating that MSCs could maintain the infection in neurogenic conditions, making this model potentially useful for prion studies [116]. Moreover, García-Mendívil and his colleagues studied the potential of ovine MSCs to be infected by natural scrapie and replicate PrP^Sc^, showing sustained levels of PrP^Sc^ postinoculation [117]. Neurons and astrocytes derived from SCs offer a promising approach to creating a cell culture model of prion infection and replication [118]. Krejciova and his colleagues demonstrated that astrocytes derived from human iPSCs could sustain the propagation of prions isolated from CJD patients’ brain samples [118]. Therefore, MSCs are suitable cell models for establishing in vitro systems to study prion infectivity and propagation.

## 7. Modulation of Hematopoietic Stem/Progenitor Cell Fate by Prion Disease

We recently reported that BM-conserved hematopoietic cells differentially express PrP^C^, and the expression of PrP^C^ gradually increases depending on more immature hematopoietic cells [119] (Figure 4).

Hematopoietic SCs (HSCs, phenotypically defined as CD150+CD48-Lineage^−^Sca-1^+^c-Kit^+^ cells) express approximately 65% of PrP^C^ protein, which is higher than the expression in Lin^−^Sca-1^+^c-Kit^+^ (LSK) cells (~48%), Lin^−^Sca-1^−^c-Kit^+^ cells (~18%) and Lin^−^ cells (~17%). PrP^C^-positive LSK cells and HSCs go through apoptotic cell death in the BM of ME7-infected middle-aged mice via elevating mitochondrial ROS, Annexin V, p-JNK, and Caspase 3 levels in a cell-autonomous mechanism. MSCs express around 48% of PrP^C^ protein in the BM of ME7-infected mice and display cell senescence by upregulating senescence-related markers such as SA-β-gal activity and p16 in a cell-autonomous mechanism. ME7-infection-caused senescence of MSCs renders the BM microenvironment aged. The preferentially aged BM microenvironment by ME7 infection causes both PrP^C^-positive and negative HSCs to become senescent by upregulating the levels of senescence-related factors such as mitochondrial ROS, p-p38, p16, and SA-β-gal activity through a non-cell-autonomous-manner in infected old-aged mice. The identification of hematological abnormalities in prion disease can aid in the finding of hints to improved survival in prion-infected individuals. Furthermore, the result raises concerns about the therapeutic use of BM cells from early prion-infected persons who do not have prion disease-related symptoms in cancer patients who require BM-conserved HSC transplantation.

## 8. MSCs as a Potential Therapy for Prion Diseases

The most common neurological disorders include neurodegenerative diseases and injury to the central or peripheral nervous system. Mahar and Cavalli 2018 concluded that after neuronal cells have been damaged, it is difficult for them to regain their functions [120]. MSCs have been reviewed as possible therapeutic agents for neurodegenerative disorders such as stroke [121], Alzheimer’s disease (AD) [122], Parkinson’s disease (PD) [123], sclerosis, and injuries of the spinal cord [124]. Because of the abovementioned potential activities of MSCs, MSC transplantation into brain lesions improves functional impairments and exhibits neuroprotective action. Several registered clinical trials on ClinicalTrials.gov(accessed 16th August 2023) employ MSCs to treat various neurological issues in individuals with AD, PD, and spinal cord injury [125].

Prion disease is one of the neurological disorders for which MSCs can be a promising cell-based therapy (Table 1).

Song and his colleagues conducted the first investigation of MSCs as a possible therapy for prion diseases in 2009, evaluating the potential effect of immortalized xenogeneic human BM-MSCs (hBM-MSCs) in prion-infected mice. MSCs could spread to brain lesion areas and extend the lifespan of infected mice. In the same study, hBM-MSCs differentiated into neural cells in response to prion infection lesions and produced a variety of trophic factors [126]. The mechanism of migration of MSCs to the brain lesions caused by prion propagation was elucidated, implying the involvement of CCR3, CCR5, CXCR3, and CXCR4 in MSC functions after chemotactic migration [130]. In another investigation, the same group extracted autologous compact bone-derived MSCs (CB-MSCs) from the femur and tibia to treat prion-infected mice. Using an in vitro migration experiment, the CB-MSCs migrated to brain extracts from Chandler-strain-infected animals. Moreover, MSC implantation could reduce body weight loss and increase microglial activation [127].

Hay and his colleagues reported that the exposure of glia and BV2 microglial cell line to prion infection in an in vitro model when co-culturing with AdMSCs resulted in a substantial reduction in inflammatory cytokine mRNA and markers for reactive astrocytes and activated microglia [131]. In a prion-infected mouse model, Hay and his colleagues recently evaluated the utility of intranasally administered adipose-derived MSCs (AdMSCs). When activated with tumor necrosis factor-alpha or prion-infected brain homogenates, AdMSCs promote anti-inflammatory genes and growth factors. Mice given AdMSCs had less vacuolization across the brain, and inflammasome signaling genes were downregulated in the hippocampus. AdMSC therapy altered the quantity and shape of the microglia, and animals had fewer reactive astrocytes [129].

## 9. NSCs as a Potential Therapy for Prion Diseases

NSCs are self-renewing multipotent cells in the nervous system that create neurons, astrocytes, and oligodendrocytes. NSCs can be obtained from PSCs, ESCs, or iPSCs [132]. It has been reported that transplanted NSCs are a promising therapy for neurological disorders by several mechanisms, such as releasing neurotrophic factors, inhibiting neuroinflammation, improved neuronal plasticity, and cell repair [133]. As aforementioned, the physiological form of PrP positively regulates the endogenous early stage of neurogenesis or adult neurogenesis [134]. However, increasing the expression of PrP^C^ can lead to increased conversion into PrP^Sc^, resulting in prion propagation in the brain tissue.

As a new cell culture model for prion disease, fetal NSCs, and adult multipotent progenitor cells could be the basis of a cell model for prion diseases as reviewed by Milhavet et al., 2006 [135]. Moreover, Relaño-Ginès and his colleagues demonstrated that PrP^Sc^ accumulated and replicated in NSCs isolated from prion-infected mice [136]. Such cells are potentially a cell therapy for prion diseases as well. The use of fetal NSCs (fNSCs) as a potential late-stage therapy for TSEs has been reported, demonstrating that the transplanting of fNSCs derived from prion-resistant knockout (koPrP) induces larger numbers of neurons and prolonged survival rate [137]. Another study showed the impacts of transplanting fNSCs, isolated from wildtype PrP or knockout PrP, into prion-infected animals on the development of the clinical signs. A significant result was indicated by increased incubation (20.1%) and survival times (13.6%). Furthermore, these temporal delays were linked to a decrease in the amount of astrocytes in areas around the NSC injection sites [128].

## 10. Conclusions

Despite significant advances in research in recent years, a therapy that stops or may delay the first signs of prion diseases or even reduces their course is still unavailable. Due to the several unique pathways that may cause prion diseases, treatment options for the disease are becoming increasingly diverse. Using SCs as a treatment approach is promising due to their capacity to regenerate damaged cells and improve clinical outcomes. SCs generated from BM and other SCs have been utilized as transplants to repair damaged neural cells in the brains of prion-induced mice. While certain outcomes have proved encouraging in terms of boosting mouse life, the timing of disease onset and SC transplantation is critical to achieving successful results. As a result, the usefulness of SCs has not yet been proven in prion diseases; the type of SC and the source employed in research must be considered.

## Figures and Tables

**Figure 1 cells-12-02413-f001:**
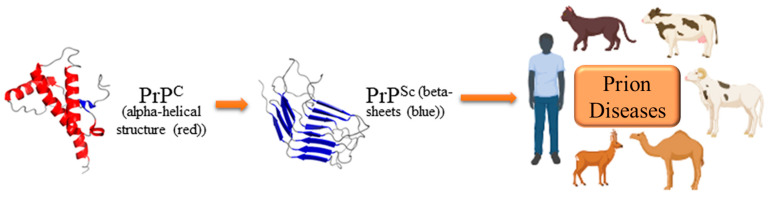
Cause and host diversity of prion diseases. Created with BioRender.com (accessed on 27 September 2023).

**Figure 2 cells-12-02413-f002:**
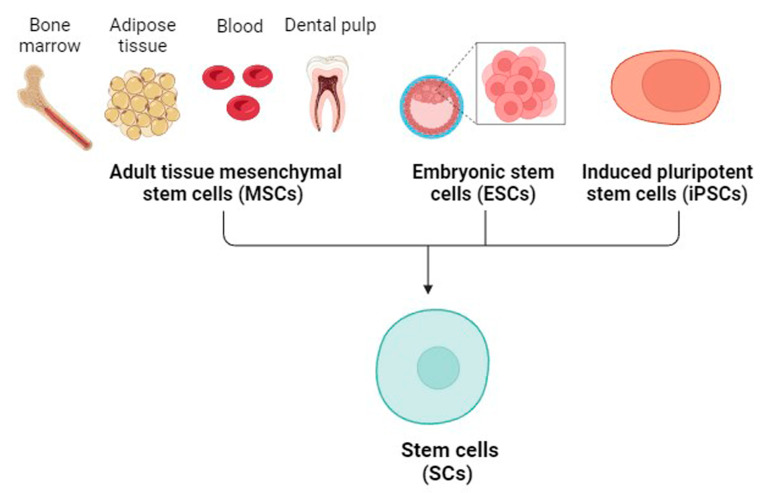
An illustration showing the different types of stem cells. Created with BioRender.com (accessed on 28 September 2023).

**Figure 3 cells-12-02413-f003:**
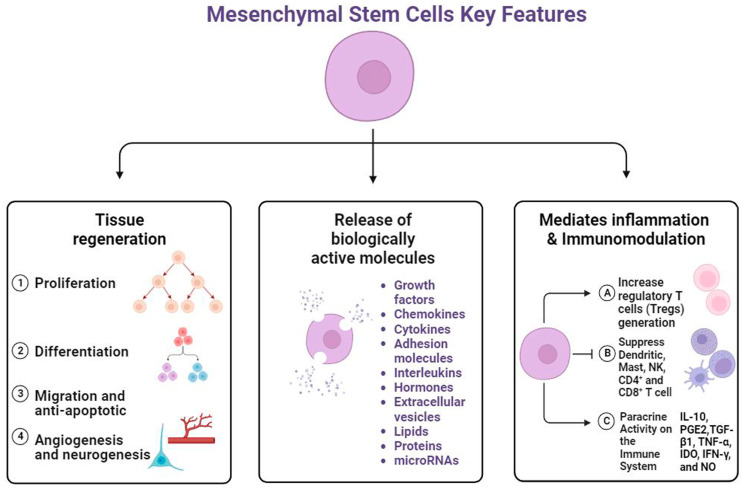
Strategies enhancing the therapeutic efficacy of mesenchymal stem cells (MSCs) include tissue regeneration, release of biologically active molecules, and mediation of immunomodulation including indoleamine 2,3-dioxygenase (IDO), prostaglandin E2 (PGE2), transforming growth factor-β (TGF-β), interleukin 10 (IL 10), and tumor necrosis factor-alpha (TNF-α), nitric oxide (NO), and interferon-gamma. Created with BioRender.com (accessed on 20 July 2023).

**Figure 4 cells-12-02413-f004:**
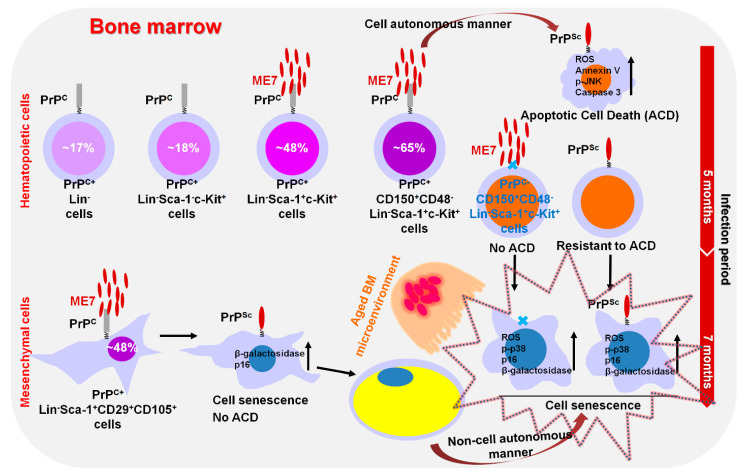
A simplified schema of the proposed model. At 5 months post-injection, PrP^C^-positive HSCs in ME7-infected middle-aged mice show cell-autonomous apoptosis. At 7 months post-injection, both PrP^C^-negative and -positive HSCs in ME7-infected old-aged mice become senescent due to selective deterioration of the BM microenvironment with aged phenotypes of increased adipogenesis and osteoclastogenesis via a non-cell-autonomous mechanism.

**Table 1 cells-12-02413-t001:** Preclinical trials of mesenchymal stem cell (MSCs) therapy for the management of prion diseases.

Preclinical Study	Cell Source	Species	Outcome	Reference
Effect of transplantation of bone marrow-derived mesenchymal stem cells on mice infected with prions.	Immortalized human bone marrow-derived MSCs	Mice infected with Obihiro/Chandler scrapie strain	Prolonged survival time Produced trophic factors and differentiated into neuronal lineages	[126]
The therapeutic effect of autologous compact bone-derived mesenchymal stem cell transplantation on prion disease.	Autologous compact bone-derived MSCs	Mice infected with Obihiro/Chandler scrapie strain	Enhanced microglial activation MSCs migrate to brain lesions	[127]
Stem cell therapy extends incubation and survival time in prion-infected mice in a time window–dependent manner.	Fetal NSCs	RML strains of mouse-adapted prions	Increased incubation (20.1%) and survival times (13.6%) Reduction in the number of astrocytes	[128]
Intranasally delivered mesenchymal stromal cells decrease glial inflammation early in prion disease.	Adipose-derived MSCs	RML strains of mouse-adapted prions	Decreased vacuolization Promoting a quiescent state in hippocampal microglia Decrease in reactive astrocytes	[129]

RML: Rocky Mountain Laboratories.

## Data Availability

Not applicable.

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
