# Peer review of "Potential Therapeutic Use of Stem Cells for Prion Diseases"

_cells, 2023, doi:10.3390/cells12192413_

Round 1

Reviewer 1 Report

This manuscript discusses prion diseases and their link to the cellular prion protein (PrPC), found on the surface of stem cells (SCs), which play a role in pluripotency and differentiation. Zayed et al review the recent papers and highlight highlight the potential of SCs as an in vitro model for prion diseases and their broader therapeutic application in neurodegenerative conditions. Researchers reading this manuscript can gain insights into the intricate biology of prion diseases and the promising role of stem cells in both disease modeling and potential treatments.

By delving into prion diseases' molecular mechanisms and the involvement of PrPC in SCs, this manuscript offers a potentially comprehensive understanding of the diseases. It not only underscores the potential of SCs as disease models but also points to their therapeutic promise in neurodegenerative disorders, suggesting a pathway for novel treatments based on regenerative medicine principles.

In terms of the timeline, this is a subject requiring a review and holds the potential for a meaningful review manuscript. However, the overall quality is very low, and rather than being a review of recent literature, the content of the text seems to be more about the author's attempt to assert their own thoughts. Therefore, the following aspects need to be re-evaluated:

1. The illustrations used throughout lack professionalism and fail to provide accurate and meaningful information (mostly can be searched by google image). In a review paper, illustrations serve as the first point of interest for readers and should be improved by experts to accurately represent the content.

2. The alignment between the text and the references is inadequate. Particularly, citations referring back to the review paper should be minimized, and if necessary, clearly stated as "inference" or "review opinion.".

3. Upon examining the papers cited in the review, it appears that recent research updates have not been well incorporated. There are barely ten references from papers published after 2022, and the selected references are insufficiently substantiated. It's unclear what the purpose of your review is, given the utilization of inadequately proven references.

Hence, the paper needs significant restructuring within the aforementioned framework, followed by a thorough review of the finer details.

N/A

Author Response

Comments and Suggestions for Authors

In terms of the timeline, this is a subject requiring a review and holds the potential for a meaningful review manuscript. However, the overall quality is very low, and rather than being a review of recent literature, the content of the text seems to be more about the author's attempt to assert their own thoughts. Therefore, the following aspects need to be re-evaluated:

Response: Thank you so much for your keen observation and kind evaluation. We appreciate your comments and concerns and have tried our best to improve the manuscript accordingly.

The molecular biology of prion disease is discussed initially in this work. Furthermore, the proposed roles of stem cells as a viable in vitro model for prion diseases and their intriguing therapeutic applications for prion diseases are highlighted. Moreover, we do our best to update recent literature related to the review.  (Note: Corrections/changes have been indicated with Red font in the revised manuscript).

  1. The illustrations used throughout lack professionalism and fail to provide accurate and meaningful information (most can be searched by google image). In a review paper, illustrations serve as the first point of interest for readers and should be improved by experts to accurately represent the content.

Response: Thank you very much for your comment. The figures and illustrations have been modified to provide high-quality figures.

  1. The alignment between the text and the references is inadequate. Particularly, citations referring back to the review paper should be minimized, and if necessary, clearly stated as "inference" or "review opinion”.

Response: Thank you for your suggestion. We completely agree with the reviewer. However, reviews that provide overviews to enhance the understanding of prion diseases and the potential of SCs as prion models and therapeutic approaches are not only possible but also suitable for maintaining a reasonably concise list of references and avoiding unnecessary duplications. We agree that the review article should be stated as an "inference" or "review opinion”. The manuscript has been modified accordingly, for example, on page 1, line 39.

  1. Upon examining the papers cited in the review, it appears that recent research updates have not been well incorporated. There are barely ten references from papers published after 2022, and the selected references are insufficiently substantiated. It's unclear what the purpose of your review is, given the utilization of inadequately proven references. Hence, the paper needs significant restructuring within the aforementioned framework, followed by a thorough review of the finer details.

Response: Thank you for your interesting comment. We have updated the list of references to the best of our ability. To the best of our knowledge, there are limited published articles utilizing stem cells as a cell model/potential treatment for prion disease. In this review, we offer readers an overview, and our selection of references serves as a guide for further reading and we try to cite the most appropriate references. Where necessary, we have included references to in-depth reviews on specific topics.

We are really thankful and deeply appreciate your efforts to evaluate our study. It has really helped to improve the quality of our manuscript and we sincerely hope that all the concerned queries have been addressed accordingly.       

 Thanks for your anticipation.

Reviewer 2 Report

In the review manuscript entitled Potential Therapeutic Use of Stem Cells for Prion Diseases" , Zayed and co-workers discuss the state-of-art usage of stem cells for the study and treatment of prion protein-associated pathologies. Overall, the manuscript appears well-written and well-organized, providing information on the recent advancements on the topic.

A few minor points need to be addressed before considering the manuscript for publication:

1. Figures are blurred upon zooming in. Please provide a higher quality of images.

2. In section 3.2 (lines 140-151), the authors discuss the advancement in the study of molecules inhibiting prion conversion. Although this section is correctly placed within the manuscript, it is missing critical information related to the usage of the RT-QuIC as a screening technique for anti-prion molecules, which would greatly improve the quality of the manuscript. Here are some references that can be used to expand this section:
- doi: 10.1371/journal.pone.0170266. 

- doi: 10.1007/978-1-4939-7816-8_16

- doi: 10.1080/19336896.2018.1525254.

- doi: 10.1016/j.coph.2018.10.001.

- doi: 10.1007/s12035-019-01837-w.

- doi: 10.3390/antiox11040726.

- doi: 10.1038/srep28711.

3. At the beginning of section 6 (lines 273-280), the authors claim that neither animal models nor the currently used cell culture models are of sufficient quality for the study of prion infection and transmission. The review would benefit from a more thorough explanation of why this is the case. Moreover, references are missing from this part.

4. References are missing from sentences in lines: 42-44; 78-80; 123-125; 

English language is overall fine. Minor concerns regard the end part of the abstract (line 24, where a past form is used in a paragraph written in present tense); sentence in line 65-66; 75; 104-105 could be rephrased to make it more clear to readers.

Author Response

General Comments

In the review manuscript entitled Potential Therapeutic Use of Stem Cells for Prion Diseases", Zayed and co-workers discuss the state-of-art usage of stem cells for the study and treatment of prion protein-associated pathologies. Overall, the manuscript appears well-written and well-organized, providing information on the recent advancements on the topic.

A few minor points need to be addressed before considering the manuscript for publication:

Response: Thank you so much for your keen observation and kind evaluation. We appreciate your comments and concerns and have tried our best to improve the manuscript accordingly. (Note: Corrections/changes have been indicated with Red font in the revised manuscript).

  1. Figures are blurred upon zooming in. Please provide higher-quality images.

Response: Thanks for your comment and notice. We provided higher-quality images. In addition, we modified the illustrations for Figures 1 and 2.

  1. In section 3.2 (lines 140-151), the authors discuss the advancement in the study of molecules inhibiting prion conversion. Although this section is correctly placed within the manuscript, it is missing critical information related to the usage of the RT-QuIC as a screening technique for anti-prion molecules, which would greatly improve the quality of the manuscript. Here are some references that can be used to expand this section:

- doi: 10.1371/journal.pone.0170266.

- doi: 10.1007/978-1-4939-7816-8_16

- doi: 10.1080/19336896.2018.1525254.

- doi: 10.1016/j.coph.2018.10.001.

- doi: 10.1007/s12035-019-01837-w.

- doi: 10.3390/antiox11040726.

- doi: 10.1038/srep28711.

Response: Thanks so much for providing the references and your valuable suggestion. The recommended references by the reviewer have been cited in the manuscript. In addition, the RT-QuIC technique has been outlined (section 3.2, page 4, lines 153-161).

  1. At the beginning of section 6, the authors claim that neither animal models nor the currently used cell culture models are of sufficient quality for the study of prion infection and transmission. The review would benefit from a more thorough explanation of why this is the case. Moreover, references are missing from this part.

Response: Thanks for this interesting comment. Firstly, we have provided the missing references regarding this part (section 6, page 7). In addition, we have discussed the hypotheses explaining the insufficient quality of cell culture and animal models for the study of prion biology (section 6, page 7, lines 305-323). Briefly, in addition to the cost and time consumed, anti-prion compounds discovered using mouse prion do not exhibit activity against human prion strains (Kawasaki et al. 2007). We have highlighted the inadequacy of recapitulating other features of human prion disease and the limited lifespan that prevents the development of a phenotype have been highlighted (Brandner et al. 2017). Challenges in cell-cultured models include prion strain-specific therapeutic effects, the emergence of drug-resistant prion strains following persistent treatment, and the difficulty of propagating human prions (Berry et al. 2013; Gils et al., 2015).

  1. References are missing from sentences in lines: 42-44; 78-80; 123-125

 Response: Thanks so much for your notice. The missed references have been cited in the appropriate sentences (page 1 line 47, page 2 line 82, page 3 line 129).

Comments on the Quality of English Language

The English language is overall fine. Minor concerns regard the end part of the abstract (line 24, where a past form is used in a paragraph written in present tense); sentences in lines 65-66; 75; 104-105 could be rephrased to make it more clear to readers.

Response: Thanks so much for your comment. The sentences have been rephrased (page 2, lines 69-71, line 79; page 3, lines 108-109). The manuscript has been proofread and edited by a well-known Company’s English Language Editing Service. We hope that all the edits in the revised manuscript will be acceptable.

We are really thankful and appreciate your efforts to evaluate our study; it really helped us improve the quality of our manuscript and we do hope that concerned queries have been answered accordingly. 

Thanks for your anticipation.

Reviewer 3 Report

This review entitled “Potential Therapeutic Use of Stem Cells for Prion Diseases is a well-researched and carefully illustrated summary of the understanding prion disease mechanisms and the possible therapeutic uses of stem cells (SCs) to treat these incurable conditions. The current state-of-the-art in the field is well covered with authors putting particular emphasis on using mesenchymal SCs(MSCs) and neural stem cells (NSCs), which are the self-renewing stem cells of the nervous system. They also discuss the benefits of MSC-derived extracellular vesicles (EVs), as they have a safety profile superior to that of MSCs, due to being less immunogenic and lacking cellular content. The use of fetal NSCs (fNSCs) as a potential late-stage therapy for prion diseases (PD) is also discussed.

General comments.

   This review is well organised, and the pathogenesis of the PDs is well outlined in the background section. The authors then proceed to examine important questions about the use of cell-based treatment options for PDs and a number of other disorders. Furthermore, a number of preclinical studies are outlined. In summary, this is

balanced case for using SCs to treat PDs, while being mindful of both the potential risks and undeniable promise involved.

Specific comments.

 The article is well written with a few errors – examples might include

Line 175  “a major challenge for this type of?” – the object noun is missing.

Line 236-7  “..essential role in clinical purposes” –change to “.. role for clinical purposes”   

Line 368  “..HSCs to be senescent by” –change to “..HSCs to become senescent by”

Line 407  “..the transplanting fNSCs derived”–change to “..transplanting of fNSCs

The article is well written with a few minor errors.

Author Response

General Comments

This review is well organized, and the pathogenesis of the PDs is well outlined in the background section. The authors then proceed to examine important questions about the use of cell-based treatment options for PDs and a number of other disorders. Furthermore, a number of preclinical studies are outlined. In summary, this is a balanced case for using SCs to treat PDs, while being mindful of both the potential risks and undeniable promise involved.

Response: Thank you so much for your keen observation and kind evaluation. We appreciate your comments and have tried our best to improve the manuscript accordingly. (Note: Corrections/changes have been indicated with Red font in the revised manuscript).

Specific comments.

The article is well written with a few errors – examples might include.

Line 175 “a major challenge for this type of?” – the object noun is missing.

Response: The sentence has been corrected (page 5, line 204).

Line 236-7 “..essential role in clinical purposes” –change to “.. role for clinical purposes”

Response: The sentence has been corrected (page 7, lines 267-2).

Line 368 “..HSCs to be senescent by” –change to “..HSCs to become senescent by”

Response: The sentence has been corrected (page 9, line 388).

Line 407 “..the transplanting fNSCs derived”–change to“..transplanting of fNSCs”

Response: The sentence has been corrected (page 11, line 449).

We are really thankful and appreciate your efforts to evaluate our review article; it really helped us improve the quality of our manuscript and we do hope that concerned queries have been answered correctly. 

Thanks for your anticipation.

Round 2

Reviewer 1 Report

Main concerns have been well addressed, and there is no other point for additional revision.